# Inhibition of synucleinopathic seeding by rationally designed inhibitors

**Smriti Sangwan[1,2,3†], Shruti Sahay[1,2,3†], Kevin A Murray[1,2,3], Sophie Morgan[4], Elizabeth L Guenther[1,2,3], Lin Jiang[5], Christopher K Williams[6], Harry V Vinters[6], Michel Goedert[4], David S Eisenberg[1,2,3]\***

[1]Department of Biological Chemistry, Howard Hughes Medical Institute, University of California, Los Angeles, Los Angeles, United States; [2]UCLA-DOE Institute, University of California, Los Angeles, Los Angeles, United States; [3]Molecular Biology Institute, University of California, Los Angeles, Los Angeles, United States; [4]MRC Laboratory of Molecular Biology, Cambridge, United Kingdom; [5]Department of Neurology, David Geffen School of Medicine, University of California, Los Angeles, Los Angeles, United States; [6]Department of Pathology and Laboratory Medicine, University of California, Los Angeles, Los Angeles, United States

**Abstract** Seeding, in the context of amyloid disease, is the sequential transfer of pathogenic protein aggregates from cell-to-cell within affected tissues. The structure of pathogenic seeds provides the molecular basis and enables rapid conversion of soluble protein into fibrils. To date, there are no inhibitors that specifically target seeding of Parkinson's disease (PD)-associated α-synuclein (α-syn) fibrils, in part, due to lack of information of the structural properties of pathological seeds. Here we design small peptidic inhibitors based on the atomic structure of the core of α-syn fibrils. The inhibitors prevent α-syn aggregation in vitro and in cell culture models with binding affinities of 0.5 µM to α-syn fibril seeds. The inhibitors also show efficacy in preventing seeding by human patient-derived α-syn fibrils. Our results suggest that pathogenic seeds of α-syn contain steric zippers and suggest a therapeutic approach targeted at the spread and progression that may be applicable for PD and related synucleinopathies.

**\*For correspondence:**
david@mbi.ucla.edu

[†]These authors contributed equally to this work

## Introduction

Parkinson's disease (PD), Parkinson's disease dementia (PDD), Lewy body dementia (LBD), multiple system atrophy (MSA) and several rarer diseases are together classified as 'synucleinopathies', a class of common neurodegenerative diseases characterized by the pathogenic accumulation of the 140 amino acid protein, α-synuclein (α-syn) in a subset of neuronal and glial cells. A causative link between α-syn amyloid formation and disease is supported by the findings that multiplications and missense mutations in *SNCA*, the α-syn gene, cause familial, early-onset PD and LBD (*Goedert et al., 2013*).

Crystallography, NMR and cryo-electron microscopy (cryo-EM) have provided structural details of α-syn in the amyloid state (*Tuttle et al., 2016*; *Rodriguez et al., 2015*; *Guerrero-Ferreira et al., 2018*; *Li et al., 2018a*; *Li et al., 2018b*). Limited proteolysis and NMR studies suggest that the fibril core is composed of residues 30–100 (*Miake et al., 2002*). A solid state NMR (ssNMR) and recent cryo-EM studies of recombinant α-syn fibrils revealed a Greek key-shaped α-syn topology in a compact domain extending from residue 44 to residue 97 (*Tuttle et al., 2016*; *Guerrero-Ferreira et al., 2018*; *Li et al., 2018a*; *Li et al., 2018b*). To gain atomic resolution information necessary for inhibitor design, we used x-ray and electron diffraction (*Rodriguez et al., 2015*) and focused on two short segments of α-syn. One segment is residues 68–78 termed NACore, which earlier work

(*Giasson et al., 2001*; *Periquet et al., 2007*) had suggested is critical for α-syn aggregation and pathology. Also, β-synuclein, a homolog does not contain residues 73–83 and is not found in amyloid deposits, and removal of α-syn residues 71–82 that encompass NACore has been shown to reduce the aggregation and toxicity in vitro and in a *Drosophila* model (*Giasson et al., 2001*; *Periquet et al., 2007*; *Rivers et al., 2008*). Additionally, a modification at Thr72 reduces the aggregation propensity of α-syn (*Marotta et al., 2015*). Taken together these studies suggest NACore plays a critical role in α-syn fibril formation. The other segment comprises of residues 47–56 and its atomic structure was solved by electron diffraction (*Rodriguez et al., 2015*). Both segments reveal amyloid protofilaments composed of dual β-sheet homo-steric zippers. Of interest is that both these segments also form β-sheets in the compact domain of the ssNMR and cryo-EM structures. In one of these cryoEM structures (*Li et al., 2018b*) NACore forms a homo-steric zipper as it does in the isolated segment, but in other structures both segments form hetero-zippers. In hetero-zippers each β-sheet mates with a different β-sheet as opposed to homo-zippers in which each β-sheet mates with another copy of itself. The crystallographic homo-zippers and the hetero-zippers of the longer cryoEM and ssNMR structures are not necessarily contradictory; they may reveal information about different α-syn polymorphs, consistent with biochemical data that show α-syn fibrils, which vary in morphology and cytotoxicity (*Peelaerts et al., 2015*; *Bousset et al., 2013*; *Heise et al., 2005*).

In addition to the spontaneous assembly of intracellular α-syn into amyloid fibrils, a second phenomenon that contributes to disease progression is the prion-like spread of α-syn aggregates (*Goedert et al., 2014*; *Goedert, 2015*). Staging of Lewy pathology has shown that pathology spreads over time through connected brain regions, and experimental studies have shown that small amounts of α-syn aggregates can act as seeds and induce the aggregation of the native protein (*Braak and Del Tredici, 2009*; *Braak et al., 2003*; *Masuda-Suzukake et al., 2013*; *Luk et al., 2009*; *Desplats et al., 2009*). Additional evidence for the existence of prion-like mechanisms in the human brain has come from the development of scattered Lewy pathology in fetal human midbrain neurons that were therapeutically implanted into the striata of patients with advanced PD (*Kordower et al., 2008*; *Li et al., 2008*). However, unlike canonical prions, transmission of α-syn aggregates from person to person has not been demonstrated, and different polymorphs of aggregated α-syn have not been demonstrated unambiguously in the diseased human brain.

Although α-syn amyloid formation has been extensively characterized, little headway has been made in developing therapeutics that inhibit spontaneous α-syn aggregation or reduce the prion-like spread. Among promising approaches are antibodies that sequester α-syn aggregates and small molecule stabilizers that bind α-syn monomers (*Mandler et al., 2015*; *Wrasidlo et al., 2016*). Here, we report a third class of inhibitors that bind α-syn seeds and prevent their growth and elongation. The inhibitors are designed using the atomic structure of NACore as a template. We show the efficacy of these inhibitors in preventing both fibril formation and seeding in vitro and in cell-based model systems for seeding. We test the efficacy of inhibition both on α-syn aggregates formed in the presence of inhibitors and on pre-formed α-syn aggregates, and also on α-syn aggregates extracted from autopsied brain tissues from patients with synucleinopathies.

## Results

### Rational design of α-syn aggregation inhibitors

Based on the atomic structure of NACore [68-GAVVTGVTAVA-78] as a template, we applied computational and structure-based approaches to design peptidic inhibitors. The atomic structure of NACore (*Rodriguez et al., 2015*) revealed a pair of self-complementary β-sheets forming a protofilament composed of a homo-steric zipper (*Sawaya et al., 2007*). The inhibitors were designed using Rosetta-based computational modeling to bind to the tip of the steric-zipper protofilament, thereby capping the fibrils (*Figure 1A*). We identified four candidates; S37, S61, S62 and S71 that are computed to bind favorably with one or both ends of the zipper (*Figure 1*). Our selection of these four peptidic inhibitors was based on two criteria; the inhibitors should disrupt the dual β-sheet architecture by introducing steric clashes between mating sheets, and the inhibitors should prevent extension of each β-sheet. The first round of design involved in silico testing of 9–10 residue peptides on the NACore segment, and the top candidates were then tested in vitro in aggregation assays with full-length α-syn. The most promising designs from the first round were improved by adding linker

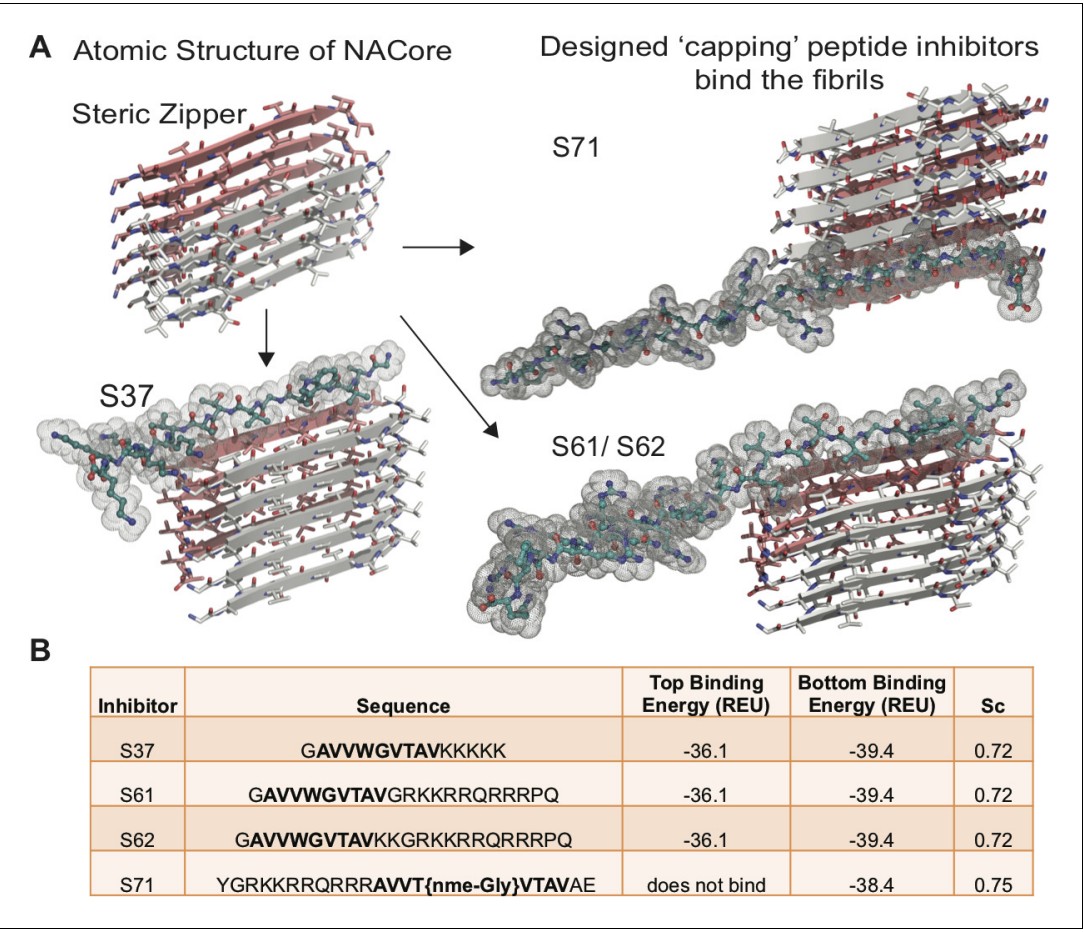

**A** Atomic Structure of NACore

Steric Zipper

Designed 'capping' peptide inhibitors bind the fibrils

S71

S37

S61/ S62

**B**

| Inhibitor | Sequence | Top Binding Energy (REU) | Bottom Binding Energy (REU) | Sc |
|---|---|---|---|---|
| S37 | G**AVVWGVTAV**KKKKK | -36.1 | -39.4 | 0.72 |
| S61 | G**AVVWGVTAV**GRKKRRQRRRPQ | -36.1 | -39.4 | 0.72 |
| S62 | G**AVVWGVTAV**KKGRKKRRQRRRPQ | -36.1 | -39.4 | 0.72 |
| S71 | YGRKKRRQRRR**AVVT{nme-Gly}VTAV**AE | does not bind | -38.4 | 0.75 |

**Figure 1.** Design of α-syn seeding inhibitors. (**A**) Structure-based design of α-syn aggregation inhibitors. The NACore protofilament is a dual β-sheet homo-steric zipper (upper left) (PDB ID: 4RIL). The inhibitors (cyan) cap the protofilament at one or both ends. S37, S61 and S62 are shown here to bind to the end with lower binding energy and S71 is shown to bind the only end to which it is predicted to bind. (**B**) Binding energies (REU, Rosetta energy units) of the four inhibitors calculated by Rosetta suggest that S37, S61 and S62 bind to both top and bottom interfaces whereas S71 is predicted to bind only to the bottom of the protofilament. Shape complementarities (Sc) of all four inhibitors with the end of the protofilament are high.

sequences and tags for improving solubility, yielding the four inhibitors. The binding energies and shape complementarity of the four inhibitors are favorable (*Figure 1B*). All the inhibitors retain most residues of the native sequence of NACore but contain one or more modified residues. Rodriguez et al. showed that a smaller 9-residue segment within NACore [69-AVVTGVTAV-77] aggregates slower than NACore and the structure is similar to NACore. In order to prevent the self-aggregation of our designed inhibitors, we used the shorter segment along with one or more modifications. S37 has a Trp replacement for Thr72 that induces a steric clash with the mating β-sheet (*Figure 1A*), and a poly-lysine tag is added at the C-terminus to produce charge-charge repulsion in the parallel β-sheet structure of NACore. S37 is predicted to bind both tips of the steric zipper fibril. S61 and S62 retain the same inhibitor sequence as S37 but instead of a poly-lysine tag, a TAT tag is added to aid solubility and prevent self-aggregation. S71 has a methylated glycine at Gly73 that weakens hydrogen bonding along the β-sheet and an additional TAT tag for solubility. Our designs ensure disruption of both homo and hetero steric zippers of NACore.

## Designed inhibitors bind to full-length α-syn fibrils with high affinity and prevent aggregation of full-length α-syn in vitro

Computational modeling predicted that our candidate peptidic inhibitors bind α-syn fibril seeds and prevent their growth and propagation. To obtain direct evidence of binding of our inhibitors to α-syn fibril seeds, we used surface plasmon resonance (SPR) (*Figure 2A*). For the SPR experiments, α-syn fibrils were immobilized on a CM5 sensor chip using standard amine coupling chemistry. For immobilization, mature α-syn fibrils were sonicated to form short fragments (*Figure 2—figure supplement 1A*) and then filtered through a 0.22-micron filter to remove large aggregates, retaining short fibrils (*Figure 2—figure supplement 1A*). We confirmed that short fibrils generated by sonication retain the ability to seed monomeric α-syn using an in vitro-seeded-aggregation assay. Aggregation assay showed that short fibrils efficiently seed the aggregation of recombinant monomeric α-syn (*Figure 2—figure supplement 1B*). We then immobilized the sonicated and filtered α-syn fibrils on a CM5 SPR chip and measured affinities of our peptide inhibitors to the fibrils. The binding assay was performed by injecting each inhibitor over two flow cells (blank control and flow cell with immobilized fibrils) at concentrations ranging from 0.3 µM to 200 µM. SPR measurements show an increase in SPR signal (response units) with increase in inhibitor concentration (*Figure 2A*). The equilibrium dissociation constant ($K_d$) was calculated by steady-state analysis, fitting the data with 1:1 binding model (*Figure 2B*). All four inhibitors bound α-syn fibrils. The inhibitor S62 showed highest binding affinity towards α-syn fibrils ($K_d$ = 0.5 µM) followed by S61 ($K_d$ = 9.28 µM), S71 ($K_d$ = 10.9 µM) and S37 ($K_d$ = 328 µM) (*Figure 2A and B*). We also tested the binding of our inhibitors towards monomeric α-syn (*Figure 2—figure supplement 2*). S37 bound with similar $K_d$ to both α-syn fibrils and monomers but the improved designs, S61, S62 and S71 showed 6–11-fold lower affinity to α-syn monomers (*Figure 2B*). Weak binding of our inhibitors to α-syn monomers is not surprising given that α-syn monomer has been reported to interact with polycations such as poly-lysine, poly-arginine and polyethyleneimine (*Goers et al., 2003*). The binding of α-syn with these polycations is attributed to the electrostatic interactions between the negatively charged C-terminus of α-syn and the positive charge of polycations. All four inhibitors in our study have a poly-lysine tag or a TAT tag, which have several lysine and arginine residues. The weak binding of our inhibitors to monomeric α-syn may be due to electrostatic interactions between the tags on our inhibitors and the C-terminus of α-syn monomers.

We further tested the efficacy of the inhibitors in an in vitro aggregation assay. Recombinantly purified α-syn was aggregated by continuous shaking at 37°C in the presence of the inhibitors and monitored by measuring fluorescence of Thioflavin T (ThT), an amyloid binding dye. All four inhibitors prevented aggregation as signaled by at least five-fold reduction in ThT fluorescence of α-syn in presence of inhibitors (*Figure 2C*). S62 and S61 showed most potency and inhibited at equimolar concentrations while S37 and S71 showed efficacy at ten-fold higher concentrations than α-syn. Electron microscopy (EM) of the aggregated samples confirmed that in the presence of the inhibitors sparse and short fibrils are formed, whereas α-syn aggregated alone forms abundant, long and unbranched fibrils (*Figure 2D*). We also observe that presence of inhibitors results in less abundant, thick bundled fibrils (*Figure 2D*).

Together the ThT and SPR experiments suggest that the inhibitors hinder α-syn fibril formation; their inhibitory effect may be attributed to their ability to bind α-syn fibril seeds and prevent fibril growth.

## α-syn aggregates formed in the presence of inhibitors show reduced seeding in cell culture models

We tested the efficacy of the inhibitors in preventing aggregation in a cell culture model. For these assays, we used two HEK293 cell lines that stably express YFP-labeled full-length wild type (WT) α-syn and the familial mutant, A53T (*Sanders et al., 2014*). In this model system, lipofectamine-mediated transfection of 125 nM recombinant α-syn fibrils seeds the aggregation of the endogenous YFP-labeled protein, which appear as fluorescent puncta (*Figure 3—figure supplement 1A and B*). Additionally, these puncta increase in size and number over time (*Figure 3—figure supplement 1C*, left). This proliferation of puncta over time is indicative of a 'seeding' phenomenon whereby a small 'seed' of amyloid fibrils induces aggregation of the endogenous protein (*Figure 3—figure supplement 1C*). For objective quantification, we used an imaging cytometer that allowed high-throughput

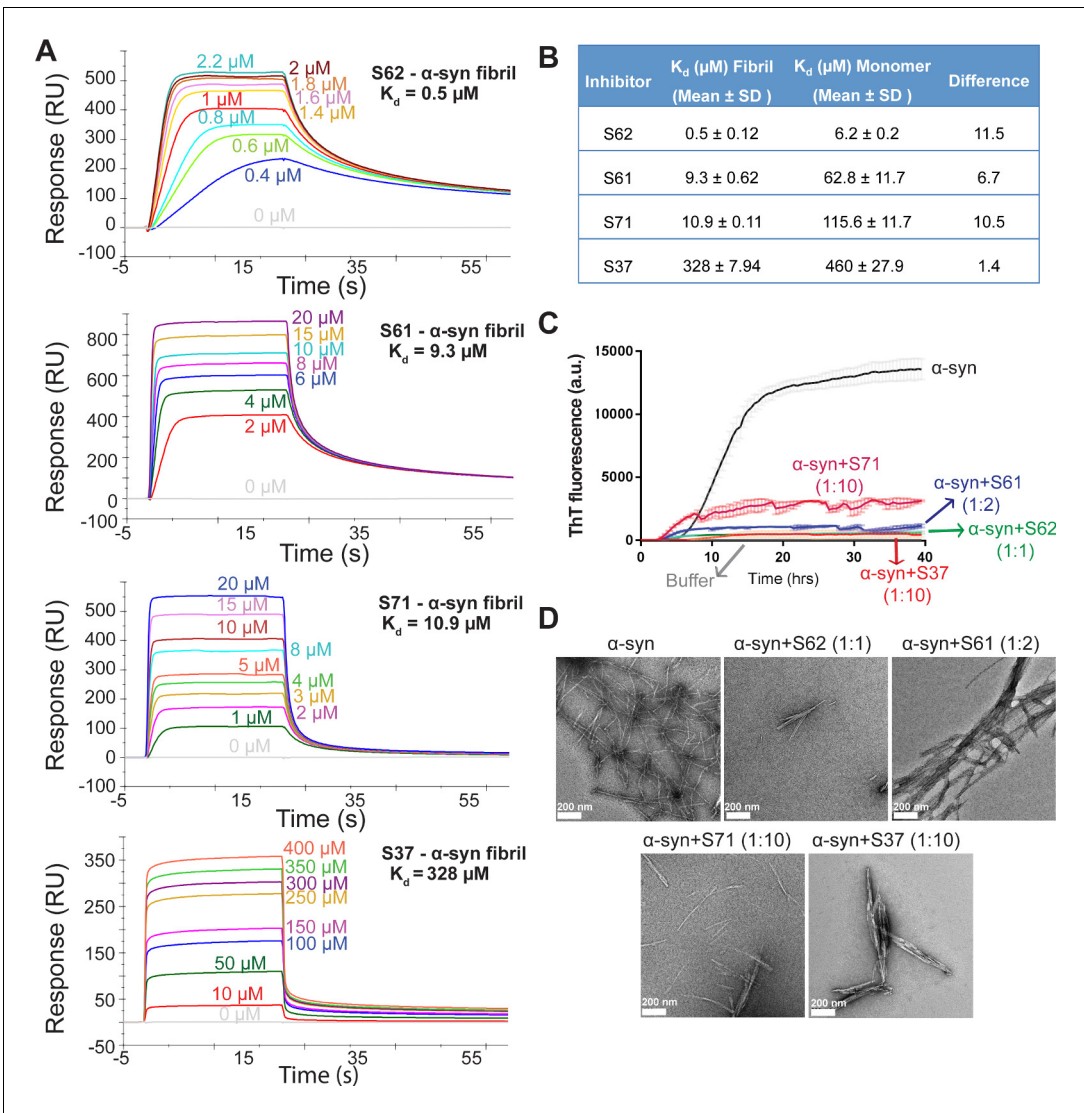

**Figure 2.** Inhibitors bind α-syn fibrils and inhibit α-syn fibril formation in vitro. (**A**) SPR measurements of the different inhibitors with α-syn fibrils. α-syn fibrils were immobilized on a CM5 sensor chip using standard amine coupling chemistry. For the binding assay, each inhibitor was injected at a flow rate of 30 μl/min at concentrations ranging from 0.5 μM to 500 μM (in running buffer, PBS, pH 7.4) at 25°C. Sensorgrams for each inhibitor showing increase in SPR signal (response units) with increase in inhibitor concentration. (**B**) The equilibrium dissociation constant (Kd) was calculated by fitting the plot of steady-state inhibitor binding levels (SPR signal at steady-state) against inhibitor concentration with 1:1 binding model. The Kd values were calculated as 0.5 μM for S62, 9.3 μM for S61, 10.9 μM for S71 and 328 μM for S37. Similar measurements with α-syn monomers (*Figure 2—figure supplement 1*) show that inhibitors bind with 1 to 11-fold lower affinity to α-syn monomers. (**C**) Thioflavin T assay to measure α-syn aggregation and the effect of inhibitors. 50 μM α-syn was aggregated in the presence of different inhibitors (α-syn:inhibitor molar ratios); S62 (1:1), S61 (1:2) and S71 (1:10) and S37 (1:10). All four inhibitors decreased ThT fluorescence indicative of inhibition of aggregation. Each curve is an average of 3 data sets and error bars represent standard deviations. (**D**) Electron micrographs of α-syn aggregated with and without the different inhibitors. Sparse fibrils are seen in the presence of inhibitors compared to α-syn aggregated alone. In the presence of inhibitors, α-syn forms short, bundled fibrils with thick morphology compared to thin long fibrils in case of α-syn aggregated alone. Scale 200 nm.

The online version of this article includes the following figure supplement(s) for figure 2:

**Figure supplement 1.** Sonicated and filtered α-syn fibrils can seed α-syn aggregation in vitro.

**Figure supplement 2.** SPR measurements of the different inhibitors with α-syn monomers (α-syn M).

screening of cells and automated counting of puncta formed over time (*Figure 3—figure supplement 1A*). In this experiment, α-syn was aggregated in the presence of 10-, 5-, 2-fold molar excess and equimolar concentrations of each inhibitor (*Figure 3—figure supplement 2*). The mixture was transfected in cells at a final concentration of 125 nM and puncta formation was visualized and counted (*Figure 3A*). We observed a dose-dependent increase in efficacy of all four inhibitors in reducing seeding in both WT and A53T α-syn expressing cells (*Figure 3B*). S62 caused significant reduction in puncta when present at equimolar concentrations, whereas S61 was effective at 2-fold molar excess or more in both cell lines. S71 and S37 showed efficacy in reducing cell seeding in both

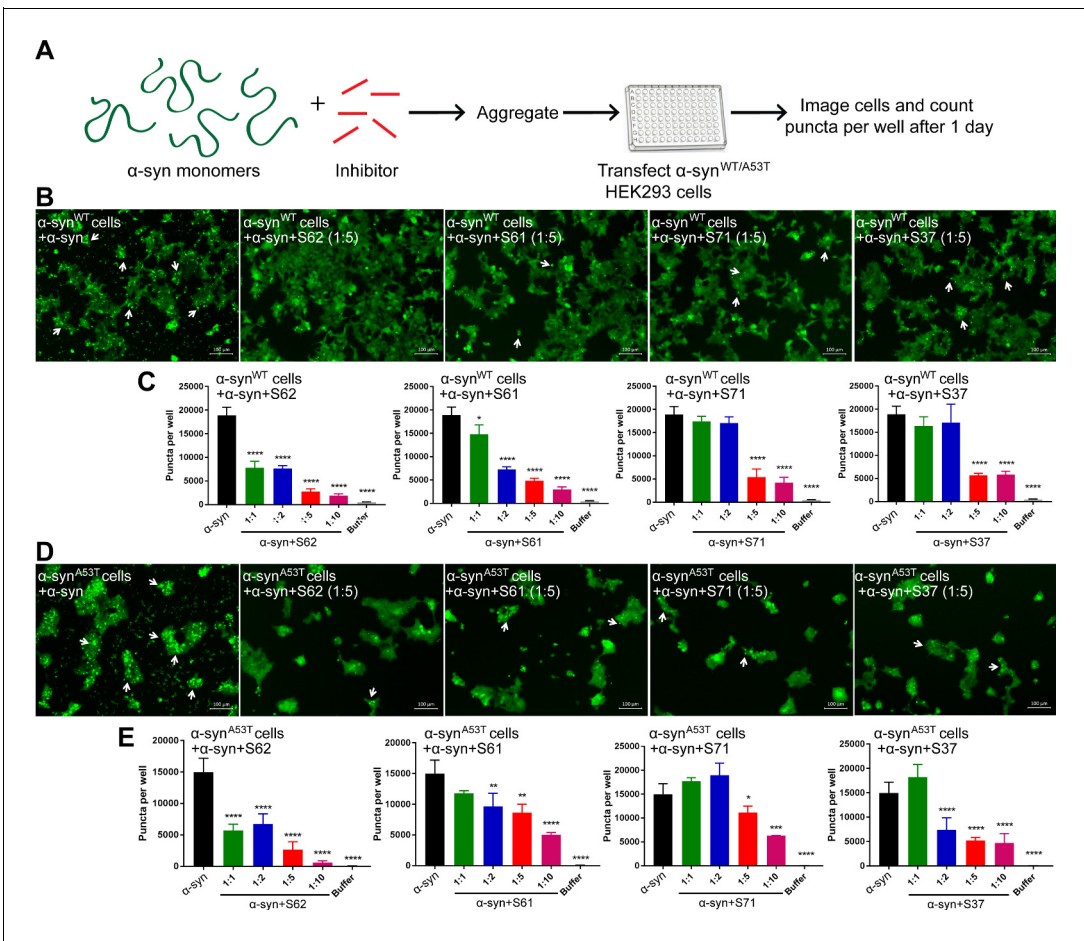

**Figure 3.** α-syn aggregates formed in the presence of inhibitors show reduced seeding ability in cells. (**A**) Experimental design of cell culture seeding assay. 50 µM recombinant α-syn monomers (green) were aggregated (by agitation at 37˚C for 2 days) in presence of various amounts of inhibitors (red). Then HEK293 cells expressing YFP labeled WT/A53T α-syn were transfected with α-syn aggregated in presence or absence of inhibitors. Cells were imaged after one day using fluorescence microscopy and the bright fluorescent puncta were counted using imaging cytometer. (**B and D**) Representative fluorescence micrographs of HEK293 cells expressing YFP-labeled WT (**B**) and A53T (**D**) α-syn transfected with aggregated α-syn in presence or absence of inhibitors. Transfection of α-syn aggregates induced endogenous α-syn to form large aggregates seen as bright puncta (white arrows). Transfection of α-syn aggregated in presence of inhibitors induced fewer puncta (white arrows) suggesting that all four inhibitors reduce the seeding ability of α-syn. Scale 100 µm. (**C and E**) Quantification of puncta formed in different conditions. With increasing concentrations of inhibitor, there is a decrease in number of puncta formed showing that the inhibitors decrease α-syn seeding ability in a dose-dependent manner. Results shown as Mean + SD (n = 3) of technical replicates. Statistical significance was analyzed by two-way ANOVA. (**p<0.01,***p<0.001, ****p<0.0001).

The online version of this article includes the following figure supplement(s) for figure 3:

**Figure supplement 1.** Cell culture seeding assay.

**Figure supplement 2.** Inhibitors reduce α-syn aggregation in vitro in a dose-dependent manner.

cell lines only at 5- and 10-fold molar excess. These results are also consistent with our SPR binding studies. S62 had the highest affinity for α-syn fibrils followed by S61 and then S71 and S37. Further, ThT fluorescence and electron micrographs showing fibril formation profile of α-syn aggregated in presence of different molar excess concentrations of inhibitors corroborate our cell seeding results (*Figure 3—figure supplement 2*). That is, S62 was most effective in inhibiting α-syn aggregation in vitro (*Figure 3—figure supplement 2A*). S62 causes about 25–65-fold decrease in ThT fluorescence of α-syn during aggregation. Electron micrographs of these samples show very sparse fibrils in the presence of equimolar and 2-fold molar excess of S62, and no fibrils are observed at 5- and 10-fold molar excess of S62 (*Figure 3—figure supplement 2A*). This corroborates our cell seeding data; α-syn aggregated in presence of equimolar and 2-fold molar excess of S62 have significantly reduced cell seeding and α-syn aggregated in presence of 5- and 10-fold molar excess of S62 lose most of their seeding capacity. Further, S61 is less potent compared to S62 in inhibiting α-syn aggregation in vitro. S61 causes 8–15-fold decrease in ThT fluorescence compared to 25–65-fold decrease in ThT fluorescence caused by S62 (*Figure 3—figure supplement 2A and B*). We observe fibrils at equimolar concentration of S61; the abundance of fibrils decreases with increase in concentration of S61 (2-, 5- and 10-fold molar excess) (*Figure 3—figure supplement 2B*). Thus, S61 is most effective in reducing cell seeding at 2-, 5- and 10-fold molar excess. S71 and S37 are less effective in inhibiting α-syn aggregation, showing a significant decrease in ThT fluorescence and number of fibrillar aggregates only at 5- and 10-fold molar excess (*Figure 3—figure supplement 2C and D*), which is consistent with their efficacy in cell seeding assay. Thus, our results from binding studies, in vitro aggregation studies and the cell seeding assay correlate.

## The inhibitors do not self-aggregate or seed α-syn aggregation in cell culture

We performed a control study to test the aggregation propensity of the inhibitors by themselves (*Figure 4*). Each inhibitor was aggregated at 500 μM concentration with continuous agitation at 37°C for two days. Fibril formation was monitored by measuring ThT fluorescence. α-syn used as a positive control showed a time dependent increase in ThT fluorescence indicative of fibril formation but none of the inhibitors showed increase in ThT fluorescence over a period of two days (*Figure 4A*). All samples were also observed by EM (*Figure 4B*). Abundant fibrils were present in α-syn incubated for two days. No fibrils were observed in any of the inhibitor samples. Only sparse, small dark structures, which may be uranyl acetate precipitates were observed in case of inhibitors incubated for two days (*Figure 4B*). The aggregated inhibitors were then tested for their ability to seed in HEK293 cells. The cell culture seeding assay showed that α-syn fibril transfected cells had abundant puncta (shown by white arrows), whereas aggregated inhibitor transfected cells did not have any puncta (*Figure 4C and D*).

## Scrambling the binding motif sequence results in loss of inhibitory effect of our designed peptides

To test the sequence specificity of our designed inhibitors we created a scrambled peptide (SP) by scrambling the binding motif sequence of S61, keeping its cell penetration tag intact. We then tested the efficacy of our SP (as a negative control) in inhibiting α-syn aggregation in vitro and cell seeding. We observed that SP did not inhibit α-syn aggregation in vitro (*Figure 4—figure supplement 1A*) and did not reduce seeding in our cell culture model (*Figure 4—figure supplement 1B and C*). This shows that scrambling the binding motif sequence results in loss of inhibitory effect of our designed peptides, and hence that the inhibitor sequence is specific for binding.

## Co-transfection of α-syn fibrils with the inhibitors prevents seeding by fibrils in cell culture

We tested the efficacy of the inhibitors to prevent seeding of pre-formed α-syn fibrils in the cell culture model (*Figure 5A*). Pre-formed α-syn fibrils were incubated with different concentrations of inhibitors at room temperature (RT) to allow binding. The mixture was then transfected in HEK293 cells expressing YFP-labelled WT/A53T α-syn (*Figure 5A*). Cells were imaged using fluorescence microscopy and the number of puncta formed was counted using imaging cytometer after one day. Two (S62 and S61) out of the four inhibitors cause reduction in puncta in both cell lines (*Figure 5B*

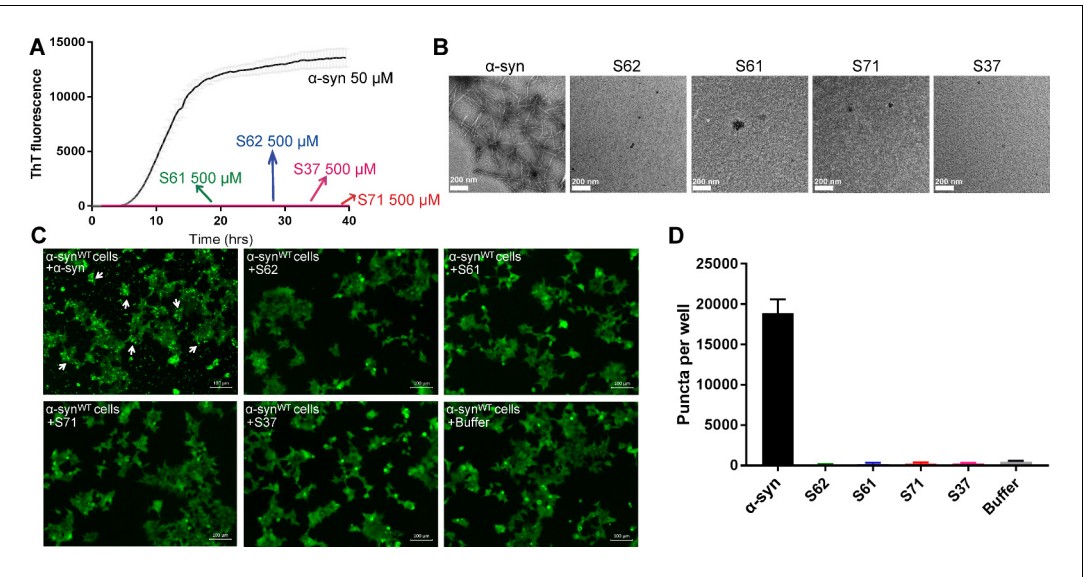

**Figure 4.** Inhibitors do not aggregate by themselves and do not seed α-syn aggregation in cells. (**A**) Inhibitors, S61, S62, S37 and S71 incubated alone (at 500 μM concentration) did not show any increase in ThT fluorescence over a time period of 2 days whereas α-syn showed a time dependent increase in fluorescence indicative of fibril formation. Each curve represents average of 3 data sets and error bars represent standard deviation. (**B**) Electron micrographs of ThT assay samples at the end of 2 days of incubation. Sparse, small amorphous aggregates are observed for any of the inhibitors, whereas abundant fibrils are observed for α-syn. Scale 200 nm. (**C**) Fluorescence micrographs of WT α-syn expressing HEK293 cells transfected with α-syn/inhibitor incubated under shaking conditions for 2 days. Cells were imaged after 1 day of transfection. Inhibitor transfected cells did not have any puncta, whereas α-syn transfected cells had abundant puncta (shown by white arrows). Scale 100 um. (**D**) Quantification of puncta formed in different conditions. In agreement with the visual findings, the quantification of puncta shows that inhibitor/buffer transfected cells did not have puncta whereas α-syn fibrils cause abundant puncta formation.

The online version of this article includes the following figure supplement(s) for figure 4:

**Figure supplement 1.** Scrambling the inhibitor sequence abolishes its inhibitory effect on α-syn aggregation and seeding.

*and D*). The quantification of the puncta formed shown as bar graphs in *Figure 5C and D* corroborates the visual findings. With increase in concentration of inhibitors S62 and S61, we observe a decrease in number of puncta formed suggesting that the inhibitors decrease α-syn seeding ability in a dose-dependent manner (*Figure 5C and D*). Notably, 25–100-fold molar excess of inhibitors was required to prevent seeding by pre-formed fibrils in the cell culture assay (*Figure 5*) as opposed to maximum 10-fold molar excess in case of cell seeding with α-syn aggregated in presence of inhibitors (*Figure 3*). We would expect higher amounts of inhibitors for efficacy against pre-formed fibrils to off-set the effects of primary nucleation, secondary nucleation and inhibitor dissociation from fibrils. Secondary nucleation, which may occur by the sides of the fibrils would create additional fibrils and thus require more inhibitor than just fibril tip-mediated primary nucleation. Incidentally, inhibitors targeted against Tau, an Alzheimer's -associated amyloid protein also show efficacy at 25–100-fold molar excess concentrations (*Seidler et al., 2018*) suggesting that secondary nucleation is a common mechanism of aggregation of amyloid proteins.

In our binding studies we found that S62 and S61 bound α-syn fibrils with higher affinity compared to S71 and S37. We suggest that higher potency of inhibitors S62 and S61 to prevent cell seeding by pre-formed α-syn fibrils may be attributed to their higher affinity for α-syn fibrils. In contrast, the inhibitors S71 and S37, which have lower affinity for α-syn fibrils do not significantly affect cell seeding ability of α-syn fibrils. Further, we performed EM analysis of pre-formed α-syn fibrils incubated with each inhibitor at RT for 3–4 hr (*Figure 5—figure supplement 1*). The inhibitors S62 and S61 which reduce the seeding ability of pre-formed α-syn fibrils also alter the morphology of the fibrils. Incubation of pre-formed α-syn fibrils with inhibitors S62 and S61 results in lateral

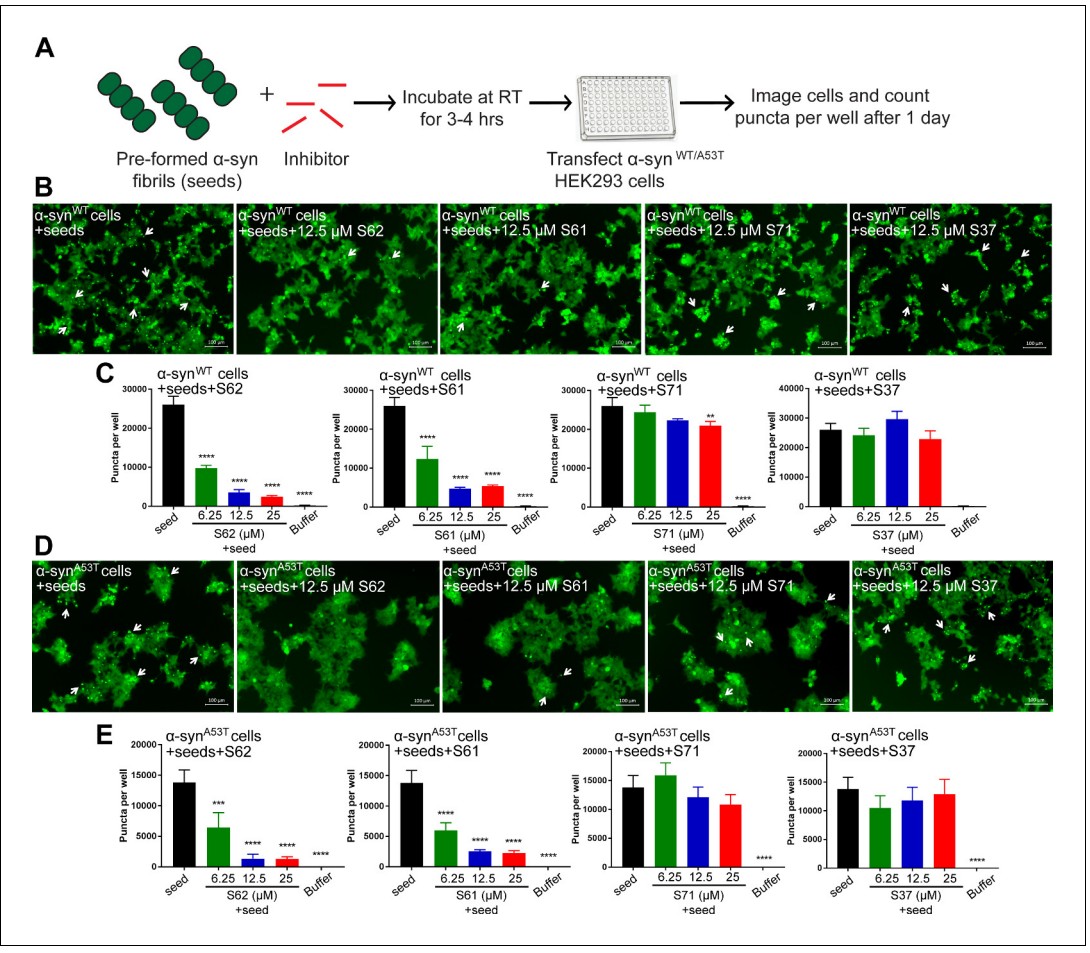

**Figure 5.** Inhibitors reduce the seeding ability of α-syn fibrils in cells. (**A**) Experimental design of cell culture seeding assay. Pre-formed α-syn fibrils (green) were incubated with various amounts of inhibitors (red) for 3–4 hr to allow binding. The mixture was transfected in HEK293 cells expressing YFP-labelled WT/A53T α-syn. After 1 day of transfection, cells were imaged using fluorescence microscopy and the bright fluorescent puncta were counted using imaging cytometer (**B and D**) Representative fluorescence micrographs of HEK293 cells expressing YFP labeled WT (**B**) and A53T (**D**) α-syn transfected with seeds and seeds pre-incubated with inhibitors. Transfection of α-syn seeds induced endogenous α-syn to form large aggregates seen as bright puncta (white arrows). Transfection of α-syn seeds pre-incubated with inhibitors S62 and S61 induced fewer puncta (white arrows) in both WT and A53T α-syn expressing HEK293 cells but S37 and S71 do not have a significant effect on puncta formation. Scale 100 μm. (**C and E**) Quantification of puncta formed in different conditions. In agreement to the visual findings the quantification of puncta shows that S62 and S61 significantly reduce the number of puncta in both cell lines. This means that pre-incubation of α-syn fibrils with inhibitors (S62 and S61) that have higher binding affinity for α-syn fibrils reduces cell seeding ability. In contrast, the inhibitors S71 and S37, which have lower affinity for α-syn fibrils do not have a significant effect on cell seeding ability of α-syn fibrils. Results shown as Mean + SD (n = 3) of technical replicates. Statistical significance was analyzed by two-way ANOVA. (**p<0.01,***p<0.001, ****p<0.0001).

The online version of this article includes the following figure supplement(s) for figure 5:

**Figure supplement 1.** Incubation with inhibitors causes morphological changes in α-syn fibrils.

association of fibrils to form thick, bundled morphologies (*Figure 5—figure supplement 1*). On the other hand, the inhibitors S71 and S37, which do not significantly affect cell seeding ability of α-syn fibrils do not affect the morphology of α-syn fibrils (*Figure 5—figure supplement 1*). Hence, our EM results are consistent with our cell seeding results. The reason for the lower seeding ability of bundled thick morphologies of α-syn fibrils is not yet clear. One possibility is that the inhibitors binds to

multiple seeds simultaneously causing their lateral association. High-resolution structural studies are required to understand the relation of structure of fibrils to their seeding ability.

## Inhibitors prevent seeding by MSA human brain tissue- derived seed in cell culture

We tested the ability of fibrils extracted from two different MSA subjects and one age-matched control subject (*Figure 6A*) to seed in our HEK293 cell culture model. The tissues obtained were from substantia nigra region of the brain of each subject (*Figure 6A*). We extracted insoluble protein aggregates using previously published protocol that included precipitation with phosphotungstate anion (PTA) and the ionic detergent sarkosyl (*Woerman et al., 2015*; *Prusiner et al., 2015*). Western blot analysis using α-syn-specific antibody showed that α-syn is present in extracts from both MSA brain samples and the control brain sample (*Figure 6B*). EM analysis of these extracts showed fibrillar structures in extracts from both MSA samples, whereas no fibril-like structures were found in control brain extracts (*Figure 6C* and *Figure 6—figure supplement 1*). Next, we transfected these

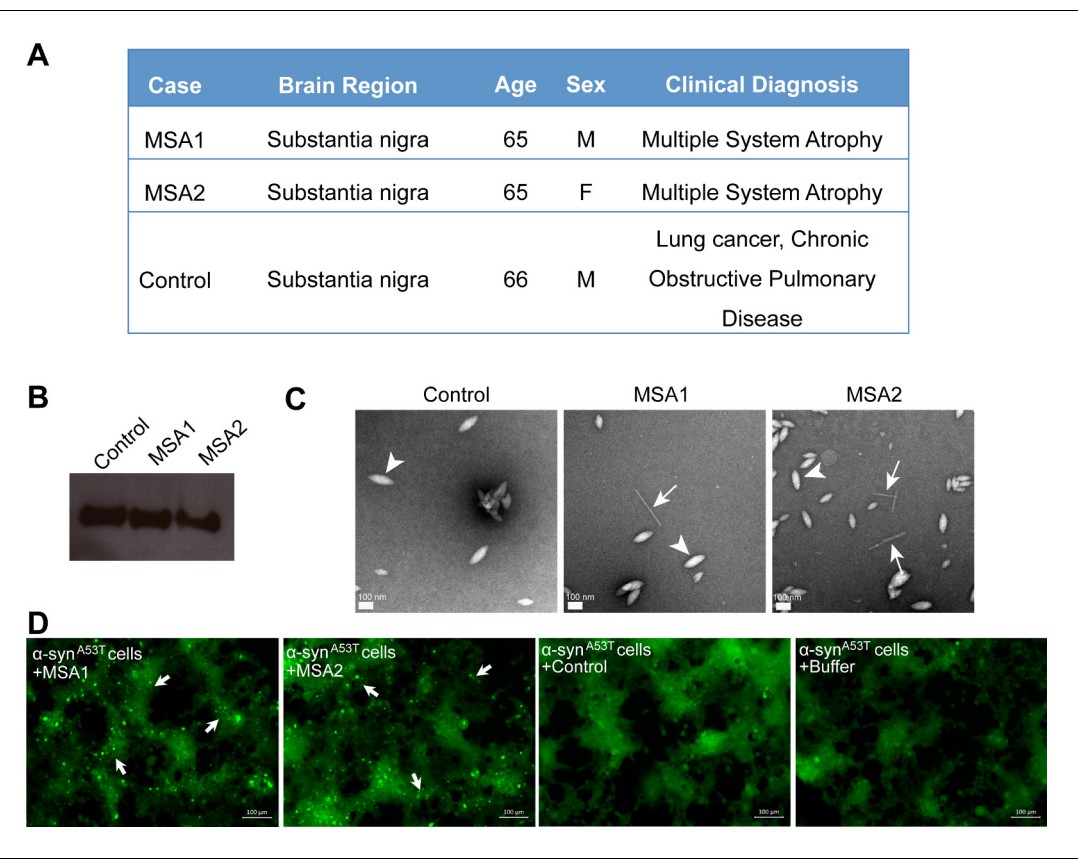

**Figure 6.** MSA derived α-syn fibrils seed α-syn aggregation in cell culture model. (**A**) Clinical information of human tissues used in this study. (**B**) Western blot of extracts from MSA1, MSA2 and control brain samples. All three samples show positive reactivity against α-syn. (**C**) Electron micrographs of extracts from MSA1, MSA2 and control brain samples. Fibrillar structures (white arrows) were found in MSA1 and MSA2 samples but not in control sample. Large amorphous structures (white arrowheads) are seen in all three samples. Scale 100 nm. (**D**) Fluorescence micrographs of HEK293 cells expressing A53T α-syn transfected with extracts from MSA1, MSA2 and control brain samples. Images were taken 7 days after transfection. Robust seeding was observed in MSA1 and MSA2 transfected cells (puncta shown by white arrows), whereas no seeding was observed in buffer or control transfected cells. Scale 100 μm.

The online version of this article includes the following figure supplement(s) for figure 6:

**Figure supplement 1.** MSA brain extracts contain fibrils.

**Figure supplement 2.** MSA derived α-syn aggregates seed α-syn aggregation in HEK293 cells expressing YFP-labeled WT/A53T α-syn.

extracts from each brain sample in HEK293 cells expressing YFP labeled α-syn. We observed robust seeding in HEK293 cells expressing A53T α-syn seven days after transfection with extracts from each of the MSA brain samples but low levels of seeding in WT α-syn expressing HEK293 cells (*Figure 6—figure supplement 2* and 6D). Transfection with buffer or extracts from control brain samples does not cause seeding in cells (*Figure 6—figure supplement 2*). Seeding by MSA derived extracts and not by control extracts suggests that the seeding is not a non-specific effect of other cellular elements present in the brain extracts but is a specific effect of α-syn aggregates present in the MSA brain samples.

We tested the efficacy of the inhibitors in preventing cell seeding by MSA derived α-syn fibrils (*Figure 7A*). For these experiments, we utilized A53T α-syn-expressing HEK293 cells, which showed robust seeding. We incubated extracts from MSA1 and MSA2 samples with each inhibitor at RT for 3–4 hr to allow binding. The mixture was then transfected in HEK293 cells. Seven days post transfection, cells were imaged using fluorescence microscopy and the puncta formed were counted using imaging cytometer. All four inhibitors significantly reduce cell seeding by both MSA1 and MSA2 samples (*Figure 7B, C, D and E*). Quantification of puncta formed in each condition showed that S62 was the most potent inhibitor followed by S71, S37 and S61 (*Figure 7D and E*). Although the

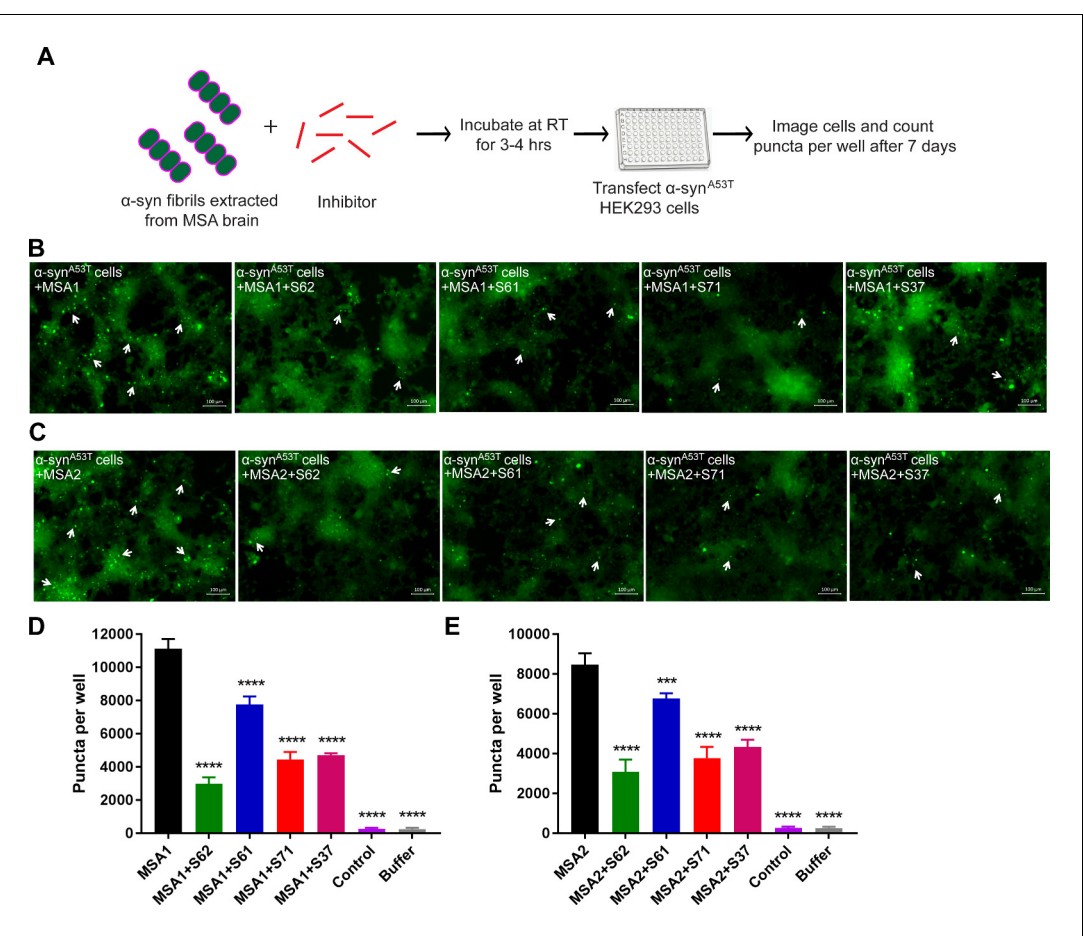

**Figure 7.** Inhibitors reduce the seeding by MSA derived α-syn fibrils. (**A**) Experimental design of cell culture seeding assay showing MSA derived α-syn fibrils (green) were incubated with each inhibitor (red) for 3–4 hr to allow binding. Then HEK293 cells expressing YFP-labelled A53T α-syn were transfected with this mixture. After 7 days of transfection cells were imaged using fluorescence microscope and the bright fluorescent puncta were counted using imaging cytometer. (**B and C**) Representative fluorescence micrographs of the cells show that all the four inhibitors cause reduction in puncta (shown by white arrows) both in MSA1 (**B**) and MSA2 (**C**) transfected cells. Scale 100 µm. (**D and E**) Quantification of puncta in different conditions. All four inhibitors significantly reduce seeding by both MSA1 and MSA2 samples. Results shown as Mean + SD (n = 3) of technical replicates. Statistical significance was analyzed by two-way ANOVA. ($**p < 0.01, ***p < 0.001, ****p < 0.0001$).

inhibitors display variability in activity, all four of our designs interact favorably with MSA derived α-syn fibrils and reduce their seeding in this cell culture model. Our designed inhibitors are effective not only against recombinant α-syn fibrils but also against patient derived fibrils demonstrating the clinical relevance of our designs.

## Discussion

We describe a structure-based approach to halt α-syn aggregation and spreading. Our approach is based on the hypothesis that the atomic structure of the NACore β-sheet is preserved in α-syn seeds, either in a homo-steric zipper or hetero-steric zipper, and that it seeds endogenous α-syn into a zipper conformation. Using the atomic structure of NACore, we developed inhibitors that hinder fibril growth and propagation and tested their efficacy in vitro and in cell culture. The inhibitors are optimized to cap the ends of α-syn fibrils, preventing further addition of monomers. We used the software Rosetta to design peptide sequences that interact favorably with the NACore segment. The energy function of Rosetta describes electrostatic interactions, hydrogen bonding and van der Waals forces, among other terms to assess binding energy. Once a specific residue has been shown to produce favorable binding in a certain position of the designed peptide, it can be fixed, while the rest of the sequence is further optimized. We performed this process of fixing and redesign iteratively until an optimal set of sequences was identified. This rational design process allowed for computationally sampling of orders of magnitude more inhibitor sequences than what was experimentally feasible to test.

To find the most effective inhibitors, nearly 100 different designs were tested experimentally using in vitro aggregation assays with sequential rounds of optimization on the inhibitor design. For example, we observed that the location of the tag on N- or C- terminus affects the efficacy of the inhibitors. Additionally, the type of modification added can also affect its efficacy. In our case, only a Trp substitution at Thr72 was effective whereas Arg substitution was not. Although the computational approach is not powerful enough to identify a single most efficient design, it can narrow our search for candidate inhibitors, which can then be refined through rational design, by adding small modifications in linker sequences and solubility tags to further improve the efficacy. Notably, S37, S61 and S62 retain the same binding motif/inhibitor sequence but show variability in their binding affinities towards recombinant α-syn fibrils as well as their potencies in inhibiting α-syn aggregation in vitro. The difference in the activities of S37, S61 and S62 may arise from different solubility and steric accessibility of the inhibitor sequence. Nevertheless, α-syn fibril binding properties of inhibitors are consistent with their efficacy in inhibiting α-syn aggregation. The inhibitors that bind α-syn fibrils with higher affinity are more potent inhibitors of α-syn aggregation confirming the specificity of our strategy.

The efficiency of capping inhibitors in preventing seeding was tested using a HEK293 cell-based assay. In this system, transfection of nanomolar amounts of α-syn seeds causes aggregation of the endogenous α-syn into puncta. These puncta display amyloidogenic properties, binding to amyloid-specific small molecules, faithfully transferred to daughter cells upon cell division and remarkable specificity. For example, α-syn fibrils can seed only α-syn protein into aggregates and not other amyloid-forming proteins such as Tau and amyloid-β (*Sanders et al., 2014*). Notably, in this system we do not observe acute cell death upon formation of puncta (*Figure 3—figure supplement 1*, right). Our inhibitors prevent puncta formation in this system with a single administration of inhibitors being effective up to 7 days. S62 and S61, which have higher binding affinity for α-syn fibrils also, are more potent inhibitors of seeding ability of α-syn fibrils. Additionally, S71 which is predicted to bind to only one of the tips of the fibril has lower affinity for α-syn fibrils compared to S62 and S61. Most notably, despite having only slightly less α-syn fibril binding affinity compared to S61, S71 displayed only marginal reduction in seeding by recombinant α-syn fibrils. The efficacy of S71 in inhibiting seeding was similar to that of S37, which has 30-fold lower α-syn fibril binding affinity compared to S71. This is potentially due to the second fibril tip that is available for fibril elongation in presence of S71. The cell culture assay does not recapitulate other features of neurodegenerative diseases, notably cellular toxicity. Nor does this assay address the roles of non-neuronal cells. Our finding of peptide inhibitors encourages future tests of their effects in animal models.

Fibrils formed by the A53T familial mutant protein are structurally similar to WT fibrils and can be inhibited by the peptidic inhibitors. Among the seven presently known missense familial mutations

associated with PD, we chose A53T because it has been shown to increase the aggregation propensity of α-syn (*Flagmeier et al., 2016*; *Li et al., 2001*; *Narhi et al., 1999*). A structural hypothesis proposed was that the mutation stabilizes the hetero-interaction of segments, 68–78 and 47–56 that form the core of α-syn fibrils (*Tuttle et al., 2016*; *Rodriguez et al., 2015*). In one cryo-EM structure, the mutation is proposed to stabilize the interaction of two protofilaments (*Guerrero-Ferreira et al., 2018*). The inhibitors were effective in preventing seeding of A53T mutant protein in cell culture model supporting the hypothesis that the structure of the familial A53T mutant fibrils is similar to WT fibrils and can be inhibited by our capping inhibitors.

Prion-like strains of amyloid proteins have been proposed and multiple strains challenge the design of inhibitors. We extracted pathological α-syn aggregates from MSA patients' brains. We observe that MSA patient-extracted α-syn fibrils cause abundant puncta formation in our cell culture model. Since A53T α-syn has higher propensity to form fibrils, we observe more aggressive seeding by MSA patient-extracted α-syn fibrils in A53T α-syn expressing HEK cells compared to WT α-syn expressing HEK cells. Further, all four inhibitors significantly reduce the cell seeding ability of MSA patient-extracted α-syn fibrils. The four inhibitors varied in efficiency against different seeds. For example, S62, S37 and S71 were more effective in reducing seeding by MSA-derived α-syn fibrils than S61. MSA samples may contain a strain/mixture of strains of α-syn fibrils, which are structurally distinct from recombinant α-syn fibrils. Hence, the interactions between the inhibitors and MSA derived α-syn fibrils may be distinct from those between inhibitors and recombinant α-syn fibrils. Efficacy of all four of our candidate inhibitors in reducing cell seeding by MSA-derived α-syn fibrils shows the clinical relevance of our design. Further, in the NMR structure of full-length α-syn fibrils the NACore segment was found in the core although not fully extended like the crystal structure of the homo zipper (*Tuttle et al., 2016*). It is conceivable that the NMR structure and the homo-steric zipper structure are polymorphs and the inhibitors vary in efficiency against these polymorphs. α-syn polymorphs have been shown to differ in seeding capacity, morphology and cytotoxicity. Currently, there is no diagnostic to identify the different polymorphs in human subjects or in any model system. In the absence of a diagnostic, a cocktail of different inhibitors targeting different polymorphs might be effective. Our inhibitors can also be used as a diagnostic to identify different α-syn strains.

In summary, we used a combination of computational methods and rational design to develop a line of inhibitors to prevent the spread of α-syn aggregates. Our approach was made possible by the determination of the atomic structure of the core of α-syn amyloid fibrils. This approach can be adopted for other diseases where seeding plays a role in disease progression.

## Materials and methods

### Computational design of inhibitor peptides

Computational designs were done using the FastDesign protocol in Rosetta version 3.8, according to previously described methods (*Sievers et al., 2011*). Briefly, the crystal structure of the NACore segment (PDB ID: 4RIL) was used as a design template. Crystal symmetry was used to align an extended nine-residue L-amino acid blocker peptide to the fibril structure, on both the top and bottom face of the β-sheet. Full sequence optimization of the extended peptide was performed, and rigid body orientation, backbone conformation, and side-chain packing of the blocker peptide were optimized for each sequence. Shape complementarity and buried surface area for each design were calculated using the InterfaceAnalyzer protocol in Rosetta. The designs were ranked by total binding energy, and the top-ranking peptides were synthesized and further tested.

### α-syn purification

The α-syn construct was transformed into *Escherichia coli* expression cell line BL21 (DE3) gold (Agilent Technologies). For expression, 10 ml LB + Amp (100 µg/mL) was inoculated from transformed colonies and grown overnight. 30 ml of starting culture was added to a 2 L flask containing 1 L LB + Amp (100 µg/mL) and grown for 3 hr at 37°C to $OD_{600}$ = 0.6. IPTG was then added to 0.5 mM to induce protein expression, which continued for an additional 3 hr. The bacterial pellet was collected by centrifugation at 4000 rpm for 10 mins. Cell pellet was resuspended in 15 mL/L pellet lysis buffer (100 mM Tris- HCl pH 8.0, 1 mM EDTA pH 8.0) and lysed by sonication. Crude cell lysate was clarified by centrifugation at 15000 g for 30 min at 4°C. 10 mg/ml Streptomycin was added to the

supernatant and stirred on ice for 30 mins followed by centrifugation at 15000 rpm for 30 mins. Protein was then purified by ammonium sulfate precipitation by adding 0.22 g/ml ammonium sulfate and stirred on ice for 30 mins followed by centrifugation at 15000 rpm for 30 mins. The supernatant was discarded and the pellet re-suspended in 12 mL/L pellet of 20 mM Tris pH 8.0. The solution was then dialyzed against 4 L 20 mM tris pH 8.0 overnight to remove residual ammonium sulfate. Next day, the protein was purified by HiPrep Q HP column (GE Healthcare) using buffer A (20 mM Tris pH 8.0) and buffer B (20 mM Tris pH 8.0; 0.5M NaCl) using a gradient from 0–100% buffer B over 100 mL. Fractions containing protein were collected and pooled and injected on a preparative size exclusion silica G3000 column (Tosoh Bioscience). The column buffer comprised 0.1 M sodium sulfate, 25 mM sodium phosphate, and 1 mM sodium azide, pH 6.5. The purified protein was dialyzed in 0.1 M sodium sulfate and 25 mM sodium phosphate twice for 4 hr and 12 hr to remove sodium azide. Protein fractions were collected and concentration measured by Pierce BCA protein assay (Thermo #23225).

## Inhibitor synthesis

The peptide inhibitors were commercially obtained from Genscript Inc at greater than 98% purity. The inhibitors were stored in lyophilized form in a desiccator box at −20°C. The lyophilized inhibitors were freshly dissolved in PBS pH 7.4/0.1 M sodium sulfate and 25 mM sodium phosphate buffer pH 6.5 before each experiment. Solubilized inhibitors were filtered with 0.2 μM filter and the concentration was calculated by measuring the absorbance at 280 nm.

## In vitro aggregation assay

Frozen aliquot of the purified recombinant α-syn was thawed on ice and diluted to 50 μM in 0.1 M sodium sulfate and 25 mM sodium phosphate buffer pH 6.5. In vitro aggregation was initiated by incubating 150 μl, 50 μM α-syn in presence of 20 μM Thioflavin T (ThT) (amyloid binding dye), in black Nunc 96-well optical bottom plates (Thermo Scientific). To test the effect of inhibitors on α-syn aggregation, α-syn was incubated in the presence of different α-syn:inhibitor molar ratios, 1:1, 1:2, 1:5 and 1:10. One PTFE bead (0.125 inches diameter) was added in each well for facilitating agitation and mixing. The plates were incubated for 2–3 days in a microplate reader (FLUOstar Omega, BMG Labtech) at 37°C with double orbital shaking at 600 rpm. Fluorescence measurements were recorded every 30 mins using $\lambda_{ex}$ = 444 nm, $\lambda_{em}$ = 482 nm. All samples were added in triplicate and experiments were repeated at least twice.

## Surface plasmon resonance (SPR) binding assay

SPR experiments were performed using BiacoreT200 instrument (GE Healthcare). α-syn fibrils/monomers were immobilized on a CM5 sensor chip. α-syn fibrils were prepared by incubating 500 μl of α-syn monomers at 500 μM concentration in PBS pH 7.4 in torrey pine shaker at 37°C, shaking speed 9 for 4 days. α-syn fibrils were isolated from the incubation mixture by centrifuging it at 13,000 xg, 4°C for 45 min. The supernatant was removed, and the pellet was re-dissolved in an equal volume of PBS as that of supernatant. The isolated α-syn fibrils were sonicated using a probe sonicator for 1 min at 18% amplitude with 2 s on, 5 s off pulses. The sonicated fibrils were filtered through a 0.22 μm filter to remove large aggregates. The sonicated and filtered fibrils were diluted to 5 μg/ml in 10 mM NaAc, pH three and then immobilized immediately on a CM5 sensor chip using standard amine coupling chemistry. Briefly, the carboxyl groups on the sensor surface were activated by injecting 100 μl of 0.2 M EDC and 0.05 M NHS mixture over two flow cells. The fibrils were then injected at a flow rate of 5 μl/min over one flow cell of the activated sensor surface for 900 s. The remaining activated groups in both the flow cells were blocked by injecting 120 μl of 1 M ethanolamine-HCl pH 8.5. Similar procedure was used for immobilizing α-syn monomers. Frozen aliquot of α-syn monomer was thawed on ice, diluted to 100 μg/ml in 10 mM NaAc, pH four and then immobilized immediately on a CM5 sensor chip. For the binding assay, each peptide inhibitor was injected at a flow rate of 30 μl/min over both flow cells at concentrations ranging from 0.5 μM to 500 μM (in running buffer, PBS, pH 7.4) at 25°C. For each sample the contact time and dissociation time were 30 s and 70 s, respectively. The data were processed and analyzed using Biacore T200 evaluation software 3.1. The data of flow cell with blank control was subtracted from the data of flow cell with immobilized fibrils/monomers. The equilibrium dissociation constant ($K_D$) was calculated by fitting the plot of steady-

state peptide binding levels ($R_{eq}$) against peptide concentration (C) with 1:1 binding model (*Equation 1*). All binding assays were performed thrice, and the data are represented as the mean value ± standard deviation.

$$R_{eq} = \frac{CR_{max}}{K_D + C} + RI \tag{1}$$

$R_{max}$ = Analyte binding capacity of the surface; RI = Bulk refractive index contribution in the sample.

## In vitro-seeded aggregation kinetics

Frozen aliquot of the purified recombinant α-syn was thawed on ice and diluted to 50 µM in PBS pH 7.4. The seeded fibril growth was initiated by incubating 150 µl α-syn (50 µM), ThT (20 µM) in the presence and absence of 2 µM α-syn fibril seeds (intact fibrils/sonicated fibrils/sonicated and filtered fibrils) in black Nunc 96-well optical bottom plates (Thermo Scientific). α-syn fibril seeds (2 µM) incubated alone were used as control. The plates were incubated for 4–5 days in a microplate reader (FLUOstar Omega, BMG Labtech) at 37°C with double orbital shaking at 600 rpm. Fluorescence measurements were recorded every 30 mins using $\lambda_{ex}$ = 444 nm, $\lambda_{em}$ = 482 nm.

## Transmission electron microscopy (TEM)

5 µl α-syn at 10 µM concentration was spotted on freshly glow-discharged Formvar-Carbon film 400 mesh copper grids (Electron Microscopy Sciences). After 4 min incubation, grids were rinsed with 5 µL ultrapure water and stained with 3 µl filter sterilized uranyl acetate solution (2% w/v in ultrapure water) for 4 min. Excess uranyl acetate was removed by blotting and another 3 µl of 2% uranyl acetate was spotted on the grid. After 3 min incubation, excess uranyl acetate was removed by blotting and the grids were allowed to air dry for 5 min. TEM images were taken using a FEI Tecnai T12 cryo-electron microscope at 120 kV with 15,000X magnification.

## HEK293 cell culture

HEK293 cells that stably express YFP labeled WT α-syn and A53T α-syn were a generous gift from Dr. Marc Diamond. Cell line was generated and validated in *Sanders et al. (2014)* and no additional validation was performed. Cells were grown in Dulbecco's modified Eagle's medium (Gibco) supplemented with 10% fetal bovine serum (HyClone), 1% penicillin/streptomycin (Gibco), and 1% glutamax (Gibco) in a humidified incubator in 37°C, 5% $CO_2$.

## Seeding in HEK293 cells

10000 live cells in 90 µL media were plated in 96 well black wall plate (Cat #3660) and allowed to adhere overnight. Each sample from in vitro aggregation assay (α-syn incubated alone or in presence of different concentrations of each inhibitor; α-syn monomer concentration 50 µM) were diluted in OptiMEM media (1:20) and sonicated in a Cup Horn water bath for three mins at low pulse. For co-transfection of pre-formed α-syn fibrils with inhibitors, the fibrils (α-syn monomer concentration 50 µM) were diluted in OptiMEM (1:20) and sonicated in a Cup Horn water bath for three mins at low pulse. Then, the fibrils were incubated with different concentrations of each inhibitor (inhibitor:α-syn molar ratio 25:1 to 100:1) for 3–4 hr at room temperature. Just before transfection, Lipofectamine 2000 was diluted in OptiMEM media (1 µL + 19 µL) and incubated at room temperature for five mins. Diluted lipofectamine and protein samples were then mixed 1:1 and incubated at room temperature for 20 mins and 10 µL was added to each well. Each sample was transfected at a final α-syn monomer concentration of 125 nM. All samples were added in triplicate and experiments were repeated at least twice. Images of cells in each well were acquired with an Axio Observer D1 fluorescence microscope (Zeiss) using fluorescent GFP channel at 10 X magnification.

## Measurement of intracellular puncta in cells

Puncta formation and cell growth was measured using Celigo Imaging Cell Cytometer allowing for unbiased measurement. Wells were imaged using fluorescent GFP channel and confluence was measured using Celigo analysis software. Images of entire wells were taken, and puncta were counted by ImageJ using the particle analysis plugin. Same settings were used to analyze wells of one plate

at all days. Total puncta counted in each well were normalized against the confluence and are reported as puncta per well.

## Extraction of protein aggregates from MSA and control human brain tissues

Frozen brain tissue samples from two different MSA subjects and one age-matched control subject were obtained from UCLA Brain Tumor Translational Resource (BTTR) and The Human Brain and Spinal Fluid Resource Center (NIH Neurobiobank). From the frozen brain tissue samples we extracted insoluble protein aggregates using previously published protocol that included precipitation with Phosphotungstate anion (PTA) and the ionic detergent sarkosyl (*Woerman et al., 2015*; *Prusiner et al., 2015*). Briefly, frozen tissue from each subject was homogenized in ice cold PBS (10% wt/vol) using Fisherbrand Disposable Tissue Grinders (Cat # 02-542-10). The brain homogenate was centrifuged at 1500 x g for five mins. The supernatant was combined with benzonase and sarkosyl to a final concentration of 0.5% and 2%, respectively. The mixture was incubated at 37°C in a Torrey Pine shaker at speed 9 for 2 hr. PTA, pH 7.0 was then added to a final concentration of 2%. This mixture was incubated at 37°C in a Torrey Pine shaker at speed 9 overnight. The sample was then centrifuged at 16000 x g for 30 min at room temperature. The resulting pellet was re-suspended in 2% sarkosyl and 2% PTA solution and incubated for 1 hr. The sample was centrifuged at 16000 x g for 30 min at room temperature. The resulting pellet was re-suspended in 10% of the initial starting volume of PBS.

## Western blot of extracts from MSA and control human brain tissues

The final resuspended pellet from the tissue extraction procedure was mixed 1:1 with 4X SDS gel running buffer. The mixture was boiled for 15 min in a water bath. Then 10 µL of each sample was loaded on NuPAGE 12% Bis-Tris pre-cast protein gel and run at 200 V for 45 min. The protein was then transferred to a nitrocellulose membrane using iBLOT2 Dry Blotting System. The membrane was treated with blocking solution (5% non-fat skimmed milk powder in TBST) for 1 hr at room temperature. The blot was then washed three times with TBST for 5 min each. The blot was incubated with primary antibody (anti-α-syn, BD Bioscience, Cat. 610787, 1:500 dilution in 2% non-fat skimmed milk powder in TBST) overnight at 4°C. After incubation, the blot was washed three times with TBST for 5 min each, followed by 1 hr incubation with horseradish peroxidase-conjugated secondary antibody (goat anti-mouse IgG H and L (HRP), ab6789, Abcam, 1:3000 dilution in 2% non-fat skimmed milk powder in TBST). The blot was then washed three times with TBST for 5 min each. The signal was detected using Pierce ECL Plus Western Blotting Substrate (Cat # 32132) and Carestream BIO-MAX Light Film (Cat # 868 9358).

## Seeding in HEK293 cells with extracts from MSA and control human brain tissues

10000 live cells in 90 µL media were plated in 96 well black wall plate (Cat #3660) and allowed to adhere overnight. The final resuspended pellet from the tissue extraction procedure was diluted 10-fold in OptiMEM and sonicated in a Cup Horn water bath for 4 hr. The sample was then incubated with each inhibitor (480 µM concentration) at room temperature for 3–4 hr. Just before transfection, Lipofectamine 2000 was diluted in OptiMEM media (1 µL + 19 µL) and incubated at room temperature for five mins. Diluted lipofectamine and protein samples were then mixed 1:1 and incubated at room temperature for 20 mins and 10 µL was added to each well. The final concentration of inhibitor in each well was 25 µM. All samples were added in triplicate and experiments were repeated at least twice.

### Statistical analyses

All statistical analyses were done using Graphpad Prism 7.0. Unless stated otherwise all assays were performed in triplicate (technical replicates) and repeated at least twice.

## Acknowledgements

The authors thank Dr. Marc Diamond for sharing the YFP-labeled α-syn expressing HEK293 cells. SS is supported by a UCLA graduate division Dissertation Year fellowship. We thank UCLA Brain Tumor Translational Resource and the Human Brain and Spinal Fluid Resource Center (NIH Neurobiobank) for tissue samples and NIH AG054022 and AG054022 for support.

## Additional information

### Competing interests

Michel Goedert: Reviewing editor, *eLife*. David S Eisenberg: SAB member and equity holder in ADRx, Inc. The other authors declare that no competing interests exist.

### Funding

| Funder | Grant reference number | Author |
| --- | --- | --- |
| National Institutes of Health | AG054022 | David S Eisenberg |
| National Institutes of Health | R01AG060149 | Lin Jiang |

The funders had no role in study design, data collection and interpretation, or the decision to submit the work for publication.

### Author contributions

Smriti Sangwan, Conceptualization, Data curation, Formal analysis, Investigation, Writing - original draft, Writing - review and editing; Shruti Sahay, Data curation, Formal analysis, Investigation, Writing - review and editing; Kevin A Murray, Software, Investigation, Methodology; Sophie Morgan, Elizabeth L Guenther, Investigation; Lin Jiang, Software, Methodology; Christopher K Williams, Harry V Vinters, Resources; Michel Goedert, Supervision, Writing - review and editing; David S Eisenberg, Conceptualization, Data curation, Funding acquisition, Writing - original draft, Writing - review and editing

### Author ORCIDs

Smriti Sangwan (iD) https://orcid.org/0000-0003-3772-7142
David S Eisenberg (iD) https://orcid.org/0000-0003-2432-5419

### Decision letter and Author response

Decision letter https://doi.org/10.7554/eLife.46775.sa1
Author response https://doi.org/10.7554/eLife.46775.sa2

## Additional files

### Supplementary files

• Transparent reporting form

### Data availability

All data generated or analysed during this study are included in the manuscript. A source data file has been provided for Figures 2,3,4,5 and 7.

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
