## [Decision Letter]

Thank you for submitting your work entitled "Inhibition of synucleinopathic seeding by rationally designed inhibitors" for consideration by *eLife*. Your article has been reviewed by a Senior Editor, a Reviewing Editor, and four reviewers. The reviewers have opted to remain anonymous.

As you will see, the referees raised several critical issues that would require substantial additional experimentation. While we cannot consider your study for publication at this stage, we would however be prepared to evaluate a resubmission that thoroughly addresses the referees' concerns.

Reviewer #1:

Prion-like aggregate spreading between neurons and brain regions is now widely considered as a mechanism contributing to the progression of neurodegenerative diseases such as Parkinson's disease (PD) and Amyotrophic Lateral Sclerosis.

In this manuscript, the authors report on four 15-22 residue long peptides designed to target the growing ends of α-synuclein amyloid fibrils, aggregates prevalent in PD. All the peptides have a nonapeptide binding motif in common, but differ in (solubility) tags, and thus act by the same mechanism. At high concentrations (161 – 500 microM), the peptides abolish α-synuclein amyloid formation in vitro. Surprisingly, the dissociation constants to α-synuclein fibrils show considerable variation (0.5 – 328 microM). In cell culture, the peptides impair aggregate seeding in HEK cells expressing YFP-α-synuclein for both wild-type and A53T mutant α-synuclein. The highly variant dose dependencies (for some peptides, anti-correlation with effector concentration!?), despite extremely low errors, are hard to rationalize – probably three technical replicates are not sufficient to get reasonable error estimates. Using fibrils grown with seeds from patient samples, the peptides show strong variation in efficacy in the different assays, perhaps reflecting different strain properties.

The study may be viewed as a proof-of-principle demonstration, as the pharmacological value of large peptides is questionable, considering their low potency and uncertain mode of delivery. However, smaller, chemically more stable compounds may act by a similar mechanism.

Major technical point: A negative control, such as a peptide with scrambled binding motif sequence, should be included to demonstrate the specificity of the effect.

Reviewer #2:

In the well-written manuscript by Godert, Eisenberg, and their colleagues entitled "Inhibition of synucleinopathic seeding by rationally designed inhibitors", they provide substantial evidence that it is possible to inhibit a seeded aggregation by their peptide inhibitors in vitro, designed to incorporate at the end of a cross-β sheet and cap the cross-β-sheet amyloid structure from further growth.

It is worrisome that the peptide-based inhibitors bind to Asyn monomer with affinities within an order of magnitude of the binding of the peptide-based inhibitors to amyloid fibrils. This implies that the inhibitors themselves are forming aggregates, perhaps even cross-β-sheet-based amyloid fibrils. Can the authors test this hypothesis specifically, and if this is not the explanation, at least speculate on a reasonable hypothesis for this surprising observation.

There were a lot of convincing experiments presented that the peptide-based inhibitors can prevent the fibrils extracted from patients from seeding recombinant Asyn (presumably by inhibition of primary nucleation, which would be appropriate to look into). The experiment I wanted to read about and the experiment that is highly relevant to the use of these peptide inhibitors to treat patients never came. While the authors showed that patient-derived fibrils could seed Asyn amyloidogenesis in HEK cells (primary nucleation, secondary nucleation or both?), they did not show that co-transfection of their peptide inhibitors with patient-derived Asyn fibrils (in the absence of recombinant protein) blocks in cell seeding. This is obviously a critical missing experiment, as this simulates what happens in a patient where cell-to-cell spreading is thought to be critical in the pathology of the synucleiopathies.

Reviewer #3:

In their study Sangwan and colleagues try to target the seed-mediated aggregation of α-synuclein by small peptide inhibitors. Therefore, they designed peptide inhibitors based on the atomic structure of the NACore region and finally analyze their effects on a-synuclein (a-syn) aggregation in cell-based and cell-free aggregation assays. The authors show that peptide inhibitors indeed can slow down seed-mediated a-syn aggregation in both model systems. Furthermore, they present experimental evidence that also the seeding activity of patient-derived a-syn structures can be decreased upon treatment with peptide molecules.

The paper is interesting for the scientific community and should be published in *eLife*. However, I have a few important concerns that need to be specifically addressed to improve the manuscript.

Essential revisions:

Figure 2:

Figure 2A: The maximum ThT fluorescence values in the presented experiments are highly variable (from 150 to 100,000 a.u). Furthermore, very high standard deviations are presented. This indicates that the established assay is very variable and not very reproducible. Data of independent experiments and multiple technical replicates need to be shown to be convincing. A Z´factor for the ThT assay should be presented.

To be able to compare the effects of the tested peptides identical concentrations should be systematically tested in a well-standardized ThT-based a-syn aggregation assay.

Figure 2B: The EM images are too small and the scale bars are difficult to read. Higher resolution and quality pictures need to be presents. The currently presented EM results are not conclusive.

Figure 3:

Why was S71 tested in lower concentrations compared to the other inhibitors? Why do the authors obtain higher numbers a-syn aggregates (puncta) when they apply higher concentrations of the S37 peptide? How do the results relate to the in vitro data (ThT assay and EM analysis)? Please show systematic experiments with different concentrations of the S37 peptide.

Figure 3—figure supplement 1:

The presented EM images of the human tissue preparations are not conclusive. To be conclusive immunogold-labeling of the extracted structures/fibrils with α-syn-specific antibodies has to be shown. It is key for this study to show the morphology of the isolated α-syn seeds. This may also at the end explain why certain peptides are more active in seed-mediated reactions than others.

Figure 3—figure supplement 2:

It seems obvious from the presented data the effect of patent-derived fibrils on the lag phase is minimal in the ThT aggregation assays. This result must be very worrying for the authors because α-syn fibrils in vitro clearly grow by a nucleation-dependent mechanism. The observed higher ThT values upon seed addition may be due to unspecific binding of ThT molecules to the additionally proteins. Thus, in my view it would be crucial to show that the isolated seeds indeed promote α-syn nucleation in ThT assays. A positive control with preformed synthetic α-syn fibrils also needs to be assessed.

Figure 5:

In Figure 3—figure supplement 2 the samples PD3, PD6, LBD2 and LBD3 were used to assess their "seeding-potential" in ThT assays. The same samples should also be investigated in cells. In Figure 5 the results for the samples LB1, PD1, LBD2 and LBD3 are presented. However, data should be presented for the same samples with both assays to be conclusive.

The ThT graphs in Figure 5 appear very crowded and their visual representation could be improved

Figure 6:

The variability of the ThT signals in Figure 6 is very high (see A, B and E). This needs to be improved to be fully conclusive.

At least for one active peptide inhibitor that influences the seeding activity of disease-relevant α-syn structures from patients an interaction with the prepared patient seeds should be shown.

Reviewer #4:

The manuscript by Sangwan et al., describes design of peptidic inhibitors that is based on the NACore structure that was reported by the same group. The manuscript is well-written but there are some major concerns regarding this study. Whether the inhibitors reported in the manuscript have the potential to be developed for further therapeutic use can only be known after performing further experiments. The major concerns regarding the conclusions drawn from this study are listed below.

1) It is difficult to identify the novel aspect presented in this study. Many studies have used peptide sequences based on the amyloidogenic region to prevent aggregation. It is well known that the region 68-78 forms a central portion of the core of α synuclein fibril. Analogous approaches of using same sequences for inhibiting aggregation have been used extensively for many other amyloids that are not referenced here.

2) The inhibitors are designed based on NACore structure that is not formed from the full length α synuclein protein. Based on point 1 it is not clear whether the fibril structure was indeed necessary to design the inhibitors. It is also possible that the NMR structure of full length fibrils could have provided better design for inhibitors.

3) Why were the inhibitors not tested in a neuronal cell line? Is there any specific reason for choosing a non-neuronal cell line?

4) Since the non-neuronal cell line does not exhibit cellular toxicity, any extrapolation towards Parkinson's disease or other neurodegenerative diseases is questionable. Protection from cellular toxicity would be essential to demonstrate the usefulness of the molecules.

5) The inhibitors prevent extension of fibril. Would that mean that there is an accumulation of protofilaments or oligomers? Is it possible that these protofilaments or oligomers are more toxic than fibrillar species? This comes back to the lack of toxicity data.

6) Do the inhibitory peptides aggregate by themselves?

---

## [Author Response]

Reviewer #1:

Prion-like aggregate spreading between neurons and brain regions is now widely considered as a mechanism contributing to the progression of neurodegenerative diseases such as Parkinson's disease (PD) and Amyotrophic Lateral Sclerosis.In this manuscript, the authors report on four 15-22 residue long peptides designed to target the growing ends of α-synuclein amyloid fibrils, aggregates prevalent in PD. All the peptides have a nonapeptide binding motif in common, but differ in (solubility) tags, and thus act by the same mechanism. At high concentrations (161 – 500 microM), the peptides abolish α-synuclein amyloid formation in vitro. Surprisingly, the dissociation constants to α-synuclein fibrils show considerable variation (0.5 – 328 microM).

Reviewer #1’s concise summary is accurate, but to us the variation in dissociation constant is not surprising given that the inhibitors differ in their tags. We think the tags can induce charge-charge repulsion and steric hindrance, and thus different tags will have slightly different effects.

In cell culture, the peptides impair aggregate seeding in HEK cells expressing YFP-α-synuclein for both wild-type and A53T mutant α-synuclein. The highly variant dose dependencies (for some peptides, anti-correlation with effector concentration!?), despite extremely low errors, are hard to rationalize – probably three technical replicates are not sufficient to get reasonable error estimates.

We agree that highly variant dose dependencies of the peptides in HEK293 cell seeding assay are hard to rationalize. We have repeated the cell seeding experiments in presence of a range of inhibitor concentrations (Figure 3 and Figure 5). For our previous experiments the concentration of the peptides was calculated based on their weight. For our repeat experiments we calculated the concentration of inhibitors by measuring absorbance at 280 nm to get more accurate concentration values. Our results now show dose-dependent increase in efficacy of each inhibitor in reducing cell seeding. The new results are shown in Figure 3 and Figure 5 and described in the accompanying text.

Using fibrils grown with seeds from patient samples, the peptides show strong variation in efficacy in the different assays, perhaps reflecting different strain properties.The study may be viewed as a proof-of-principle demonstration, as the pharmacological value of large peptides is questionable, considering their low potency and uncertain mode of delivery. However, smaller, chemically more stable compounds may act by a similar mechanism.

We agree that the study is a proof-of-principle demonstration. The main significance of our work is that an inhibitor designed to cap just the core segment of the fibril is sufficient to inhibit fibril formation of the full-length protein.

Major technical point: A negative control, such as a peptide with scrambled binding motif sequence, should be included to demonstrate the specificity of the effect.

We have now included a negative control, a peptide with scrambled binding motif sequence. We created the scrambled peptide(SP) by scrambling the binding motif sequence of S61, keeping its cell penetration tag sequence intact. We then tested the efficacy of our SP (as a negative control) in inhibiting α-syn aggregation in vitro and cell seeding. SP did not inhibit α-syn aggregation in vitro and also did not reduce seeding in our cell culture model (Figure 4—figure supplement 1). The loss of inhibitory effect of our designed peptides by scrambling the sequence provides evidence of the specificity of our designs. These results are shown in Figure 4—figure supplement 1 and described in the accompanying text.

Reviewer #2:

*In the well-written manuscript by Godert, Eisenberg, and their colleagues entitled "Inhibition of synucleinopathic seeding by rationally designed inhibitors", they provide substantial evidence that it is possible to inhibit a seeded aggregation by their peptide inhibitors* in vitro, designed to incorporate at the end of a cross-β sheet and cap the cross-β-sheet amyloid structure from further growth.It is worrisome that the peptide-based inhibitors bind to Asyn monomer with affinities within an order of magnitude of the binding of the peptide-based inhibitors to amyloid fibrils. This implies that the inhibitors themselves are forming aggregates, perhaps even cross-β-sheet-based amyloid fibrils. Can the authors test this hypothesis specifically, and if this is not the explanation, at least speculate on a reasonable hypothesis for this surprising observation.

We tested the aggregation propensity of the inhibitors and found that none of the inhibitors form aggregates or seed α-syn aggregation in cell culture. We incubated the inhibitors alone (in absence of recombinant α-syn protein) and we do not see any increase in ThT signal with time during incubation or fibrillar aggregates (under electron microscope) at the end of incubation period (Figure 4). We have now included these results in the manuscript. These results are shown in Figure 4 and described in the accompanying text.

Binding of the peptides to α-syn monomers is not surprising given that α-syn monomer has been reported to interact with polycations such as poly-lysine, poly-arginine, polyethyleneimine (Goers et al., (2003)). The binding of α-syn with these polycations is suggested to be attributed to the electrostatic interactions between the negatively charged C-terminus of α-syn and the positive charge of polycations. All four peptides in our study have a poly-lysine tag or a TAT tag which has several lysine and arginine residues. This may explain the binding of these peptides to α-syn monomers. We have added the above explanation in the manuscript (subsection “Designed inhibitors bind to full-length α-syn fibrils with high affinity and prevent aggregation of full-length α-syn in vitro*”*).

There were a lot of convincing experiments presented that the peptide-based inhibitors can prevent the fibrils extracted from patients from seeding recombinant Asyn (presumably by inhibition of primary nucleation, which would be appropriate to look into). The experiment I wanted to read about and the experiment that is highly relevant to the use of these peptide inhibitors to treat patients never came. While the authors showed that patient-derived fibrils could seed Asyn amyloidogenesis in HEK cells (primary nucleation, secondary nucleation or both?), they did not show that co-transfection of their peptide inhibitors with patient-derived Asyn fibrils (in the absence of recombinant protein) blocks in cell seeding. This is obviously a critical missing experiment, as this simulates what happens in a patient where cell-to-cell spreading is thought to be critical in the pathology of the synucleiopathies.

We have now performed the missing experiment: we used Multiple System Atrophy (MSA) patient-derived α-syn fibrils to transfect HEK293 cells expressing YFP-labelled WT/A53T α-syn. We observed robust seeding in HEK293 cells expressing A53T α-syn 7 days after transfection with extracts from each of the MSA brain samples (Figure 6). We tested the efficacy of the inhibitors in preventing cell seeding by MSA derived αsyn fibrils (Figure 7). We observe that co-transfection of inhibitors with MSA patient-derived α-syn fibrils blocks cell seeding. The systematic experiments and results are now discussed in the manuscript (Figure 6 and Figure 7 and accompanying text).

Reviewer #3:

In their study Sangwan and colleagues try to target the seed-mediated aggregation of α-synuclein by small peptide inhibitors. Therefore, they designed peptide inhibitors based on the atomic structure of the NACore region and finally analyze their effects on a-synuclein (a-syn) aggregation in cell-based and cell-free aggregation assays. The authors show that peptide inhibitors indeed can slow down seed-mediated a-syn aggregation in both model systems. Furthermore, they present experimental evidence that also the seeding activity of patient-derived a-syn structures can be decreased upon treatment with peptide molecules.The paper is interesting for the scientific community and should be published in eLife. However, I have a few important concerns that need to be specifically addressed to improve the manuscript.Essential revisions:Figure 2:Figure 2A: The maximum ThT fluorescence values in the presented experiments are highly variable (from 150 to 100,000 a.u). Furthermore, very high standard deviations are presented. This indicates that the established assay is very variable and not very reproducible. Data of independent experiments and multiple technical replicates need to be shown to be convincing. A Z´factor for the ThT assay should be presented.To be able to compare the effects of the tested peptides identical concentrations should be systematically tested in a well-standardized ThT-based a-syn aggregation assay.

The highly variable maximum ThT fluorescence arises because the assays were performed in two different Omega microplate readers, which have different settings. Very high standard deviations in the ThT may also be due to the presence of DMSO in the samples. Previously we dissolved our peptides in DMSO and so, we used α-syn+DMSO (Veh) as a control, which has high variability across replicates, thus high standard deviation. However, the peptides in this study have solubility tags and are easily soluble in aqueous buffer. We have repeated the in vitro ThT assays without DMSO (by dissolving peptides directly in the aqueous buffer) and we observe small standard deviations (Figure 2C). Our repeated data have small standard deviations and show significant discrimination between ThT curves of α-syn incubated alone and in presence of inhibitors. ThT fluorescence data from the repeated experiments are shown in the revised manuscript (Figure 2C, Figure 4A and Figure 3—figure supplement 2).

Figure 2B: The EM images are too small and the scale bars are difficult to read. Higher resolution and quality pictures need to be presents. The currently presented EM results are not conclusive.

We have added higher resolution, clearer electron micrographs in Figure 2D in the revised manuscript.

Figure 3:

*Why was S71 tested in lower concentrations compared to the other inhibitors? Why do the authors obtain higher numbers a-syn aggregates (puncta) when they apply higher concentrations of the S37 peptide? How do the results relate to the* in vitro *data (ThT assay and EM analysis)? Please show systematic experiments with different concentrations of the S37 peptide.*

We have repeated the cell seeding experiments with same range of concentrations for each inhibitor (Figure 3 and Figure 5). Our results now show dose-dependent increase in efficacy of each inhibitor in reducing cell seeding. Further, in our revised manuscript we have shown systematic experiments including ThT assay, EM analysis (Figure 3—figure supplement 2) and discussed how the cell seeding results relate to the in vitro data (subsection “α-syn aggregates formed in the presence of inhibitors show reduced seeding in cell culture model”).

Figure 3—figure supplement 1:The presented EM images of the human tissue preparations are not conclusive. To be conclusive immunogold-labeling of the extracted structures/fibrils with α-syn-specific antibodies has to be shown. It is key for this study to show the morphology of the isolated α-syn seeds. This may also at the end explain why certain peptides are more active in seed-mediated reactions than others.

We performed western blot analysis as well as EM analysis of extracts from MSA brain samples and age matched control brain sample. Our western blot analysis using α-syn-specific antibody showed that α-syn is present in extracts from both MSA brain samples as well as the control brain sample (Figure 6B). However, fibrillar structures were only found in MSA tissue extracts and not in the control brain extract (Figures 6C and Figure 6—figure supplement 1). Morphologically, the fibrils are short and straight. We have now included the above data in the revised manuscript (Figure 6 and accompanying text).

Why certain peptides are more active than others, is indeed an important question. It can perhaps be answered by structural studies of inhibitors bound to the fibrils, a goal for our future work. Our trials with immunogold labeling have been inconclusive.

Figure 3—figure supplement 2:It seems obvious from the presented data the effect of patent-derived fibrils on the lag phase is minimal in the ThT aggregation assays. This result must be very worrying for the authors because α-syn fibrils in vitro clearly grow by a nucleation-dependent mechanism. The observed higher ThT values upon seed addition may be due to unspecific binding of ThT molecules to the additionally proteins. Thus, in my view it would be crucial to show that the isolated seeds indeed promote α-syn nucleation in ThT assays. A positive control with preformed synthetic α-syn fibrils also needs to be assessed.

We agree that the low effect of patient-derived fibrils on lag phase in ThT assays is a concern. We think it is due to the low amount of seeds present in our extractions because we observe sparse fibrils by EM (Figure 6C). With the limited amount of tissues available, we decided to focus on obtaining seeding in cell culture. We have now performed our cell seeding assay with extracts from two MSA brain samples and an age-matched control. Seven days after transfection with extracts from patient-derived brain samples, we observed seeding in HEK293 cells expressing WT/A53T α-syn. It is noteworthy that, the time required to observe puncta (intracellular α-syn aggregates) in HEK cells transfected with patient-derived α-syn seeds is much longer (6-7 days after transfection) compared to the time required to observe puncta in HEK cells transfected with recombinant α-syn seeds (1 day after transfection). We speculate that the amount of α-syn seeds present in the patient-derived brain extracts is low and hence, does not show seeding in the ThT assay. Before the low amounts of seeds can cause any effect on aggregation, α-syn starts to aggregate on its own in the time-frame of our in vitro aggregation studies. We believe that our cell culture model is a much more sensitive system for detecting α-syn seeds compared to our in vitro ThT assay. Therefore, we have now shown the cell seeding assay with patient-derived samples in the revised manuscript (Figure 6). We have also shown that co-transfection of inhibitors with patient-derived seeds blocks seeding in HEK293 cells (Figure 7).

Figure 5:In Figure 3—figure supplement 2 the samples PD3, PD6, LBD2 and LBD3 were used to assess their "seeding-potential" in ThT assays. The same samples should also be investigated in cells. In Figure 5 the results for the samples LB1, PD1, LBD2 and LBD3 are presented. However, data should be presented for the same samples with both assays to be conclusive.The ThT graphs in Figure 5 appear very crowded and their visual representation could be improved

We have now repeated the seeding experiments and replaced previous patient-derived seeding data in the manuscript. We tried seeding experiments with LBD and PD samples but we did not observe significant seeding in our cell culture model. We could observe significant seeding only with MSA substantia nigra tissue- derived extracts. This is also in agreement with previous seeding studies reported by Prusiner et al., (2015). Hence, we performed our cell seeding and co-transfection with inhibitors assays with the 2 MSA brain samples and 1 age-matched control (Figure 6 and Figure 7).

Figure 6:The variability of the ThT signals in Figure 6 is very high (see A, B and E). This needs to be improved to be fully conclusive.At least for one active peptide inhibitor that influences the seeding activity of disease-relevant α-syn structures from patients an interaction with the prepared patient seeds should be shown.

We have now replaced the patient-derived α-syn seeding studies with our new data (see Figure 6 and Figure 7). As mentioned above, instead of ThT assays, we show seeding in cell culture and its inhibition by our inhibitors.

We agree that measuring the binding of our inhibitors to patient-derived seeds is an important experiment. Unfortunately, the crude extract protocol and low amount seeds limit its use in SPR experiments. The α-syn seeds derived from patients are extracted from the frozen brain tissue samples using previously published protocol (Prusiner et al., (2015)). The extraction protocol gives a crude extract, which consists of many other tissue-derived materials in addition to α-syn fibrils (please see electron micrographs of extracts from MSA and control brain samples shown in Figure 6C). SPR assays will be affected by non-specific interactions of the tissue extract with the SPR sensor. Thus, SPR assay will not give an accurate measurement of the binding between inhibitors and patient-derived α-syn seeds. Also, as mentioned above because the amount of α-syn seeds in patient tissue extract is already low, additional purification steps will lead to further lowering of α-syn seeds, which will not be sufficient for our binding studies. Therefore, we have not performed binding studies with patient-derived samples.

Reviewer #4:

The manuscript by Sangwan et al., describes design of peptidic inhibitors that is based on the NACore structure that was reported by the same group. The manuscript is well-written but there are some major concerns regarding this study. Whether the inhibitors reported in the manuscript have the potential to be developed for further therapeutic use can only be known after performing further experiments. The major concerns regarding the conclusions drawn from this study are listed below.1) It is difficult to identify the novel aspect presented in this study. Many studies have used peptide sequences based on the amyloidogenic region to prevent aggregation. It is well known that the region 68-78 forms a central portion of the core of α synuclein fibril. Analogous approaches of using same sequences for inhibiting aggregation have been used extensively for many other amyloids that are not referenced here.

The novel aspect of our study is the use of structure-based design of inhibitors. We used the atomic structure of the NACore segment of α-syn as a template to design α-syn peptide inhibitors. Rossetta-based computational modeling was employed to design inhibitors which bind the tip of the steric-zipper protofilament and prevent their further elongation. SPR experiments confirm that the inhibitors bind to α-syn fibrils with 1-11fold higher affinity than α-syn monomers. The fact that inhibitors designed based on the structure of a small segment of α-syn are able to inhibit the aggregation of full length α-syn in vitro and in cells suggests that our approach works. We agree there are other papers present in the literature in which peptide inhibitors inhibit aggregation of α-syn and other amyloidogenic proteins but these are not based on the atomic structure of α-syn fibrils.

2) The inhibitors are designed based on NACore structure that is not formed from the full length α synuclein protein. Based on point 1 it is not clear whether the fibril structure was indeed necessary to design the inhibitors. It is also possible that the NMR structure of full length fibrils could have provided better design for inhibitors.

The NACore structure from the crystallographic results of Rodriguez et al., (2015) has more recently been found in a nearly full length α-syn cryoEM structure (Li et al., (2018)). Which polymorph of α-syn fibrils is the most relevant in brain disease is currently unknown. But in any case, we have designed peptides which inhibit the aggregation of full length α-syn in vitro and in cells. Also, in the ssNMR-determined structure, part of NACore is at the core of the domain. It is possible that binding to this segment is sufficient to inhibit fibril elongation of that polymorph too.

3) Why were the inhibitors not tested in a neuronal cell line? Is there any specific reason for choosing a non-neuronal cell line?

We used the non-neuronal cell line HEK293 cells over-expressing YFP fused α-syn because they are a very convenient and well-established model for studying the seeding of α-syn in cells (Sanders et al., (2014)). These cells have been successfully used in several other studies involving α-syn seeding (Prusiner et al., (2015)).

4) Since the non-neuronal cell line does not exhibit cellular toxicity, any extrapolation towards Parkinson's disease or other neurodegenerative diseases is questionable. Protection from cellular toxicity would be essential to demonstrate the usefulness of the molecules.

In this work, we have exclusively focused on inhibiting seeding as recent work has shown that seeding is an important determinant of disease progression. Our results encourage future studies to test these inhibitors’ therapeutic efficacy in mouse model of Parkinson’s disease. Additionally, in Parkinson’s disease as well as prion disease, fibril formation seems to be necessary for spreading, but not directly for cytotoxicity. The nature of the toxic agents is a topic of intense investigation, but not addressed in this paper.

5) The inhibitors prevent extension of fibril. Would that mean that there is an accumulation of protofilaments or oligomers? Is it possible that these protofilaments or oligomers are more toxic than fibrillar species? This comes back to the lack of toxicity data.

We did not observe any loss of cell viability in HEK293 cells transfected with the inhibitors suggesting that the treatment with inhibitors renders α-syn in an inert conformation. It is possible that inhibitors cause an accumulation of protofilaments or oligomers but from our assays these appear to be non-toxic.

6) Do the inhibitory peptides aggregate by themselves?

The inhibitory peptides do not aggregate by themselves. We tested the aggregation of inhibitors by ThT, EM and cell seeding data as shown in Figure 4 in the revised manuscript.